# Thermodynamic Genome-Scale Metabolic Modeling of Metallodrug Resistance in Colorectal Cancer

**DOI:** 10.3390/cancers13164130

**Published:** 2021-08-17

**Authors:** Helena A. Herrmann, Mate Rusz, Dina Baier, Michael A. Jakupec, Bernhard K. Keppler, Walter Berger, Gunda Koellensperger, Jürgen Zanghellini

**Affiliations:** 1Department of Analytical Chemistry, University of Vienna, 1090 Vienna, Austria; helena.herrmann@univie.ac.at (H.A.H.); mate.rusz@univie.ac.at (M.R.); 2Institute of Inorganic Chemistry, University of Vienna, 1090 Vienna, Austria; dina.baier@univie.ac.at (D.B.); michael.jakupec@univie.ac.at (M.A.J.); bernhard.keppler@univie.ac.at (B.K.K.); 3Research Cluster Translational Cancer Therapy Research, University of Vienna and Medical University of Vienna, 1090 Vienna, Austria; walter.berger@meduniwien.ac.at; 4Institute of Cancer Research and Comprehensive Cancer Center, Medical University of Vienna, 1090 Vienna, Austria; 5Vienna Metabolomics Center (VIME), University of Vienna, 1090 Vienna, Austria; 6Research Network Chemistry Meets Microbiology, University of Vienna, 1090 Vienna, Austria

**Keywords:** omics data integration, constraint-based modeling, data normalization

## Abstract

**Simple Summary:**

Cancer, but also its treatment, can lead to a reprogramming of cellular metabolism. These changes are observable in metabolite abundances, which can be unbiasedly measured via mass spectrometry metabolomics. However, even when the metabolome changes strongly, a (mechanistic) interpretation is difficult as metabolite levels do not necessarily directly correspond to pathway activities. Here we measure the changes of the cellular metabolome in colorectal cancer cell lines sensitive and resistant to the ruthenium-based drug BOLD-100/KP1339 and the platinum-based drug oxaliplatin. We map these changes onto a cancer-specific genome-scale metabolic model, which allows us not only to compute intracellular flux distributions, but also to disentangle drug-specific effects from growth differences from differences in metabolic adaptations due to resistance. Specifically, we find that resistance to BOLD-100/KP1339 induces more extensive reprogramming than oxaliplatin, especially with respect to fatty acid and amino acid metabolism.

**Abstract:**

Background: Mass spectrometry-based metabolomics approaches provide an immense opportunity to enhance our understanding of the mechanisms that underpin the cellular reprogramming of cancers. Accurate comparative metabolic profiling of heterogeneous conditions, however, is still a challenge. Methods: Measuring both intracellular and extracellular metabolite concentrations, we constrain four instances of a thermodynamic genome-scale metabolic model of the HCT116 colorectal carcinoma cell line to compare the metabolic flux profiles of cells that are either sensitive or resistant to ruthenium- or platinum-based treatments with BOLD-100/KP1339 and oxaliplatin, respectively. Results: Normalizing according to growth rate and normalizing resistant cells according to their respective sensitive controls, we are able to dissect metabolic responses specific to the drug and to the resistance states. We find the normalization steps to be crucial in the interpretation of the metabolomics data and show that the metabolic reprogramming in resistant cells is limited to a select number of pathways. Conclusions: Here, we elucidate the key importance of normalization steps in the interpretation of metabolomics data, allowing us to uncover drug-specific metabolic reprogramming during acquired metal-drug resistance.

## 1. Background

A reprogramming of metabolism is a hallmark of multiple diseases, including cancer [1]. Changes in glucose, amino acid, lipid, and cholesterol metabolism, for example, have all been associated with aberrant metabolic phenotypes observed in cancers [2]. Resulting differences in metabolism between healthy and cancerous cells hold the potential for selectively targeting cancerous cells through pharmacological and dietary interventions. As such, understanding the extent to which metabolic reprogramming occurs in different cancer cells is a fundamental requirement for better treatment options. However, not only malignant transformation, but also therapy response on drug resistance acquisition might be paralleled or even driven by metabolic changes in the malignant cells [3,4]. In case of acquired therapy resistance, dissection of the respective metabolic alterations and mechanism on a larger scale are only at the beginning and have not found widespread application yet.

In silico methods have the potential to integrate existing experimental data and to generate new hypotheses about the underlying mechanisms associated with metabolic reprogramming. Genome-scale metabolic models (GSMMs), which capture the known biochemical reactions of a given system, have previously been applied in various cancer studies [5,6], such as the investigation of metabolic heterogeneity [7] and have led to the discovery of new drug targets and biomarkers [8,9,10,11,12]. There are, however, areas of cancer research where GSMMs have not yet been applied due to a lack of available experimental data. For example, GSMMs have not yet been used extensively to study acquired drug resistance against different drug classes in different cancer cell types. Acquired therapy resistance is considered a major obstacle for curative systemic cancer treatment at progressed stages and also affects the success of anticancer metal drugs [13,14,15,16].

Metal-based drug treatments involving oxaliplatin are a standard therapy for colorectal cancer, the third most commonly diagnosed cancer [17,18,19,20]. Drug resistance, however, has been reported to develop in nearly all patients with colorectal cancer; even when using modern targeted and immunotherapy options, chemotherapy remains a major part of the colorectal cancer treatment regimen [14,15]. Platinum-based drugs are still prescribed in different lines of systemic cancer treatment in diverse tumor types and patient cohorts [21]. Although platinum-based anticancer drugs like oxaliplatin are widely-used, intrinsic and acquired resistances remain a crucial impediment in the treatment of colorectal cancer.

Acquired resistance against platinum drugs is thought to be mainly based on elevated DNA-repair mechanisms, detoxification, evading apoptosis and autophagy [22]. However, there is an increasing amount of evidence that metabolic alterations might play a pivotal role as well [23]. The clinically-investigated ruthenium-based anticancer drug BOLD-100/KP1339 has shown promising results with regard to colorectal cancer treatment [24]. BOLD-100/KP1339 (sodium trans-[tetrachloridobis (1H-indazole) ruthenate(III)]) is a prodrug [25] displaying preferential activation by reduction in the hypoxic milieu of solid tumors and does not primarily target DNA [26] and metabolic alterations are expected to be relevant. Unlike BOLD-100/KP1339, which is still under investigation for clinical applications, oxaliplatin is an already widely-applied, clinical cancer treatment. As a result, the body of literature addressing oxaliplatin resistance is notably larger than that of BOLD-100/KP1339 resistance. Nonetheless, the extent to which metallodrug resistance results in an altered metabolic profile remains poorly understood for both drug treatments and has not yet been compared. As such, it is not yet known whether metabolic reprogramming during resistance development against anticancer compounds with differenct metal centers and activity parameters are comparable or drug-specific.

Metabolomics aims to directly measure metabolite abundance from a global and unbiased perspective and has the potential to not only detect metabolic alterations but to discover diagnostic and prognostic markers and to generate hypotheses that can be validated with genetic experiments [27]. Recent progress in targeted and untargeted metabolomics approaches have resulted in a wide-ranging toolkit for studying metabolic phenotypes in terms of cellular concentrations. Mass spectrometry-based metabolomics approaches can be used for the metabolic profiling of drug-treatment responses in cancer cell lines [28,29].

While metabolomics studies provide an effective interrogation window for the cellular changes that occur in response to a change in conditions, they do not necessarily provide mechanistic insights into the reprogramming of metabolism. Metabolite pools do not inform about pathway activity, ergo corresponding metabolic fluxes are sometimes measured. Measuring metabolic fluxes, however, also suffers from several practical limitations. For example, a prolonged time for peripheral pathways to reach isotopic steady-state, the fact that simple linear pathways can only be investigated with non-stationary labelling or an increased number of samples, and the complex data analyses required for nonstationary labelling experiments often hinder a successful and comprehensive application of isotopic labelling methods [30].

Recent trends in metabolomics have shown it is always possible to measure more metabolites at more time points and to analyse the obtained results in combination with other ‘omics data sets [31,32,33,34]. While multi-omics have allowed for the identification of numerous regulatory mechanisms in cancer [35,36], their integration with fluxomics is required to gain a holistic understanding of metabolic reprogramming. To understand the mechanisms that underpin a potential reprogramming of metabolism during resistance development, observed changes in metabolite concentrations need to be placed in the context of changes in metabolic flux. GSMMs provide a platform for doing so [37]. Multiple techniques for integrating omics data sets into GSMMs have been developed [11,38,39,40,41]. While expression data sets are often used to generate system-specific models [8,42], metabolomics and proteomics data sets are used to constrain the solution space of the generated models [38,41].

Typically, constraint-based modelling (CBM) is employed to study GSMMs and to explore metabolic phenotypes in the form of steady-state fluxes [43,44]. Flux balance analysis (FBA), for example, uses linear optimization techniques to model the fluxome of GSMMs [see Orth et al. [45] for a review]. FBA, however, can lead to the prediction of thermodynamically infeasible flux solutions. Thermodynamic flux analysis (TFA) imposes additional constraints on stoichiometric models to ensure thermodynamically valid fluxes and provides a framework for integrating metabolomics data into GSMMs [46,47]; extracellular metabolite data are used to constrain the directionality of exchange reactions of the model and intracellular metabolite data can be used to constrain reactions in the model. Both intra- and extracellular metabolite data have previously been integrated into system-specific metabolic models to draw physiological conclusions about cancerous and healthy cells [48,49,50,51].

In this work, we integrate experimentally determined absolute concentrations of intracellular metabolites and medium-based metabolites and growth rates of the colorectal cancer cell-line HCT116 into a cell-line specific, thermodynamic, genome-scale metabolic model (GSMM). We consider two different models of acquired resistance in colon cancer: oxaliplatin resistant (OxR) and BOLD-100/KP1339 resistant (RuR) HCT116 cells as well as their sensitive controls to generate four model instances. To identify metabolic differences between resistant and sensitive cells, we normalize the calculated flux values by their representative growth rates. As oxaliplatin and BOLD-100 are prepared in different solvents (water versus dimethylsulfoxid (DMSO)), OxR and RuR cells were grown in the same media, but RuR and its respective control were exposed to a low DMSO background equivalent to the drugs’ stock solution solvent. To account for metabolic differences that are the results of a difference in solvent, we further normalized the results obtained for the resistant cells by their sensitive controls. Eliminating both differences in growth rate and solvent background, we are able to draw drug-specific conclusions about the metabolic changes that occur upon resistance. As a result, we are able to identify specific changes in flux that are the direct result of an acquired resistance to either oxaliplatin or BOLD-100/KP1339 treatment.

## 2. Materials and Methods

### 2.1. Cell Culture

HCT116 colon cancer cells were generously provided by Dr. Vogelstein from John Hopkins University, Baltimore. Cells were cultured in McCoy’s medium (Sigma Aldrich, Burlington, MA, USA) supplemented with 10% fetal calf serum (FCS; PAA, Linz, Austria) and 2 mM glutamine (Sigma Aldrich). Cells were selected for acquired drug resistance over several months via exposure to increasing concentrations of oxaliplatin or BOLD-100/KP1339 followed by drug-free recovery phases. Finally, the oxaliplatin-resistant HCT116 (OxR) cells were selected with 5 μM of oxaliplatin [52,53] for 24 h and BOLD-100/KP1339-resistant (RuR) cells with 200 μM of BOLD-100/KP1339 for 72 h in two-week-intervals. All cultures were grown under standard cell culture conditions and checked for *Mycoplasma* contamination.

### 2.2. Cell Viability Assay

Cells were seeded at densities of $3.5\times 10^{4}$ cells/well in 96-well microtiter plates and allowed to adhere overnight. Cells were exposed to indicated concentrations of the respective drugs for 72 h. Cell viability was determined using the 3-(4,5-dimethylthiazol-2-yl)-2,5-diphenyltetrazolium bromide (MTT) assay (EZ4U, Biomedica, Vienna, Austria) following the manufacturer’s recommendations.

### 2.3. Metabolomics Experiment

HCT116 cells, HCT116 cells with acquired oxaliplatin resistance, and HCT116 cells with acquired BOLD-100/KP1339 resistance were seeded (N=6 for each respectively) as $2\times 10^{5}$ cells/well in 12-well plate format in 1 mL McCoy’s medium (Sigma Aldrich) supplemented with 2 mM glutamine and 10% FCS. After overnight growth, wells were supplemented with 1 mL fresh medium each. HCT116 with acquired BOLD-100/KP1339 resistance and its sensitive control contained the same medium with 0.5% dimethyl sulfoxide (DMSO) used as BOLD-100 solvent. Twenty-four hours after supplementing with additional medium, cells were still not confluent. At this point, the medium was removed and cells were washed three times with 2 mL PBS (37 ${}^{\circ }C$) and snap frozen with liquid nitrogen.

### 2.4. Metabolomics Sample Preparation

The samples were randomized at the stages of the experiment including seeding, sample preparation and extraction as well as LC-MS measurement sequence. Extraction and measurement of the metabolites were based on a protocol described elsewhere [54]. Briefly, the protocol comprised cell scraping and extraction with 180 μL cold 80% methanol containing 5 mM N-ethylmaleimide (dissolved in 10 mM ammonium-formate at pH 7) with 20 μL fully ${}^{13}$C-labeled internal standard, ISOtopic solutions (Vienna, Austria). After a centrifugation step (14,000 rcf, 4 ${}^{\circ }C$, 10 min) cell extracts were directly measured with high-resolution OrbiTrap mass spectrometer.

### 2.5. LC-MS Analysis of Metabolites

The quantification of metabolites was based on Schwaiger et al. [55] and the LC-MS gradient was adapted and shortened to suit metabolites as described elsewhere [56]. Full mass scan data was acquired both in positive and negative ion mode.

### 2.6. LC-MS Analysis of Coenzymes

The analysis of free coenzyme A (CoA), acetyl-coenzyme A, palmitoyl-coenzyme A, malonyl-coenzyme (A below LOD) was carried out in an additional measurement series of the same samples and on the same instrumental setup but with a dedicated LC-MS method. The same separation was used with the same gradient and eluents, but flushing of the column started at 6 min instead of 7 min, shortening the total measurement time from 15 min to 14 min. The OrbiTrap MS settings were changed with regard to the mass range to 750–1100 m/z, the capillary temperature was lowered from 280 ${}^{\circ }C$ to 200 ${}^{\circ }C$ to reduce in-source fragmentation and the S-lense RF-level was increased from 30 to 60.

### 2.7. Determination of Total Protein Content

The applied extraction and centrifugation resulted in a pellet containing the high-molecular fraction and non-polar metabolites. This pellet was dissolved in 0.2 M NaOH overnight, diluted 1:10 in the same NaOH solution and determined for total protein content with the Thermo Micro BCA kit, according to manufacturer’s instructions.

### 2.8. Data Analysis of Metabolomics Measurement

Targeted analysis of the data was done with Skyline 20.2 (available at https://skyline.ms/project/home/software/Skyline/begin.view, accessed on 12 August 2021) extracting the [M-H]${}^{-}$ and [M+H]${}^{+}$ ions with 5 ppm mass tolerance. The absolute concentrations relied on the external calibration with internal standardization. The compounds were standardized compound-specifically where possible and class-specifically when the U${}^{13}$C equivalent was not reliably available or by U${}^{13}$C-glutamate if neither of the aforementioned were available.

Metabolites with technical repeatability (relative standard deviation) above 30% were removed from the dataset. This was based on the repeated injection and measurement of a pooled quality control sample. Furthermore, metabolites which had mean concentration below the determined lowest limit of quantification (LOQ) according to the validation of the LC-MS method described in [55] were removed.

Datasets were combined by joining the metabolite data acquired in both positive and negative mode, as well as coenzyme data in the negative acquisition mode. A further calibration was measured in positive mode for several carnitines (propionyl-carnitine, O-acetyl-carnitine, propionyl-carnitine, palmitoyl-carnitine) with the method for metabolites, since these compounds were not contained in our original calibration mixture. Also the calibration row for coenzymes was prepared freshly in this mixture to avoid degradation by storage. The external calibration of the different coenzymes (coenzyme A, acetyl coenzyme A, malonyl coenzyme A, palmitoyl coenzyme A) was measured in negative mode. For every primary thiol in the dataset (coenzyme-A, glutathione, cysteine, etc.) its N-ethyl maleimide adduct was used for quantification after it was made sure that the conversion was quantitative.

### 2.9. Measurement of Extracellular Metabolite Concentrations

$10^{5}$ HCT116 cells as well as HCT116 cells with acquired oxaliplatin resistance and HCT116 cells with BOLD-100/KP1339 resistance were seeded (N=4 for each respectively) into 12-well plate (StarLab) with 2 mL McCoy’s 5A medium (Sigma-Aldrich) containing 10% FCS (BioWest) and 4 mM glutamine. Also in the case of the sensitive HCT116 cells and the BOLD-100/KP1339-resistant cells the experiment was run with and without 0.5% DMSO. 100 μL were collected from the starting medium at the beginning of the experiment, and directly from the wells 24 h, 48 h and 72 h after seeding. Also, a cell free experiment was run to determine the contribution of abiotic glutamine decay.

### 2.10. Determination of Dry-Weight for the Cell Lines

The cell dry weight measurements of HCT116 sensitive (Sen), oxaliplatin-resistant (OxR) and BOLD-100/KP1339-resistant (RuR) cells were carried out (N=4 for each respectively) as described by Széliová et al. [57].

### 2.11. HCT116-Specific Genome-Scale Metabolic Model

Robinson et al. [58] provide the latest consensus GSMM of human metabolism called Human1. The authors used Human1 to generate cell-line specific models using gene essentiality data from previous CRISPR knockout screens [59]. Using the tINIT algorithm [42] and RNA-Seq data from HCT116 colorectal carcinoma cells they select reactions from Human1 associated with moderately and highly expressed genes to build a cell-line specific model for HCT116. We obtained the model from the authors, removed the enzyme constraints and added a further seven exchange reactions to the model to account for the excretion or uptake of cis-aconitate, fumarate, isocitrate, malate, sarcosine, succinate and xanthine that we observed in our measured time-course of the medium composition. We use this updated model for all our analyses presented here. The model is available at https://github.com/HAHerrmann/Hct116_DrugRes/blob/master/Models/Colon_Combined.xml (accessed on 12 August 2021).

Growth rates (Appendix A) and exchange rates (Appendix A) were fitted as described in Széliová et al. [60]. In short, we fitted an exponential model to estimate the initial concentration, $X_{0}$, and the growth rate, μ. The fitted growth rate and the initial biomass, $B_{0}$, were then used to calculate the specific exchange rates for all of the measured medium-based metabolites across all time points, i.e., 0, 24, 48, and 72 h. $B_{0}$ was calculated from the fitted $X_{0}$ and the experimentally determined dry mass per cell (Appendix A). The fitting was done in Python (Version 3.7.9) using the optimize function in scipy (Version 1.5.2) with parameters soft_l1 for the loss function and f_scale=0.3 for outlier detection. The propagated error of the growth rate measurement (Appendix A) and the dry mass per cell (Appendix A) was calculated and used to set upper and lower bounds. The obtained growth and exchange rates were used to constrain the respective import and export reactions of the model. Flux constraints were set such that the applied upper and lower bounds accounted for the relative standard error or the propagated error of the measurement. We further constrained the directionality of uptake and excretion rates of 50 metabolites, using HCT116 cell line specific data obtained by Jain et al. [61]. The “blood pool” reactions were removed from the model because we did not consider in vivo conditions. Instead, we allowed for an unconstrained influx of stearate, palmitate, oleate, linolenate, linoleate, and arachidonate. These fatty acids have previously been shown to make up the majority of lipids present in fetal calf serum [62,63] which was used as a growth medium supplement.

All applied model constraints are based on data obtained during the exponential growth phase and as such, all model results are specific to this growth phase.

### 2.12. Thermodynamic Metabolic Modeling

The pyTFA package [47], https://github.com/EPFL-LCSB/pytfa, accessed on 12 August 2021, formulates thermodynamic flux analysis (TFA) of GSMM as a mixed-integer linear programming problem that incorporates metabolite concentrations as thermodynamic constraints into a traditional flux balance analysis (FBA) model. Masid et al. [49] have recently constructed an extensive thermodynamic database containing the thermodynamic information for compounds, reactions and compartments in human metabolism; this includes the Gibbs free energy formation of compounds and the associated error estimation, the pH, ionic strength and membrane potentials. Using Biopython (Version 1.78) we annotated the GSMM with SEED identifiers which allowed us to match the information in the GSMM to the thermodynamic database of Masid et al. [49]. Absolute metabolite concentrations were normalized according to total protein content. Normalized values were used to constrain condition-dependent GSMM instances. Between 61 and 68 metabolites were experimentally constrained in each model instance. Where no experimental data was available, metabolite concentrations were set to the default range of $10^{-12}$ to 0.1 mol per total protein. This allowed us to achieve a thermodynamic coverage of 89% of the compounds and to estimate the Gibb’s free energy for 20% of the reactions. Using a parsimonious FBA (pFBA) that maximizes a linear objective while minimizing the total sum of fluxes [64], we calculated the minimum total sum of fluxes and set this as an additional constraint to our linear model prior to performing a Flux Variability Analysis (FVA) on the thermodynamic model, here referred to as TFVA. TFVA applies the same constraints as TFA but instead of returning a single feasible solution, the lowest and highest possible flux value for each reaction is returned [65]. Because pFBA does not necessarily return a unique solution when two alternative pathways with the same total sum of fluxes exist, we chose to implement a parsimonious TFVA (pTFVA) to compare different model instances to one another. Upon parallelizing the existing TFVA implementation in pyTFA for an improved run time, we ran a pTFVA for different instances of the HCT116 cell-lines specific GSMM. Flux analyses were done in Python (Version 3.7.9) using cobrapy (Version 0.19.0) [66].

### 2.13. Data Processing and Flux Normalization

We constrained four different instances of the HCT116 model: oxaliplatin-resistant cells (OxR) and their sensitive parental counterpart (HCT116) and BOLD-100/KP1339-resistant cells (RuR) and their sensitive parental counterpart in a DMSO-based medium (HCT116-DMSO). Model instances were constrained using the condition-specific exchange fluxes (Appendix A) and growth rate (Appendix A). All blocked reactions were removed using the find_blocked_reactions in cobrapy (Version 0.19.0) with default parameters, resulting in a model with 4530 reactions and 4492 degrees of freedom. Upon calculating flux values for each model instance using pTFVA as described, we divided each set of flux values by the outgoing flux to biomass production of that model instance, effectively normalizing for difference in growth. We checked for reactions for which both the upper and the lower bound differed by at least 15%. Furthermore, we feature-scaled all flux values to lie between 0 and 1 and divided the flux values obtained in the drug-resistant instances by the corresponding flux values obtained for their respective controls. Having thus normalized for differences in the medium composition, we were able to compare the flux profiles of the two metallodrug resistances to another, again checking for which reactions both the upper and lower bounds differed by at least 15%.

## 3. Results

### 3.1. Differences in Metabolite Concentrations May Not Correlate to Changes in Flux

To investigate the metabolic changes associated with metallo-resistance in colorectal cancer, we compared the metabolic profiles of resistant and sensitive cells. Using the HCT116 colorectal cancer cell line, cells with resistance to either oxaliplatin (OxR) or BOLD-100/KP1339 (RuR) were compared to their sensitive counterparts. The two acquired resistence models are largely independent of one another: while OxR cells show moderate cross-resistance for the ruthenium-based drug, RuR cells display no cross-resistance and remain sensitive to oxaliplatin treatment (Appendix A). This implies a difference in the molecular basis of resistance between the two models. OxR cells and their parental sensitive counterparts were grown in a standard medium, while RuR cells and their parental sensitive counterparts were grown in the same medium but with a low solvent-background (DMSO) as outlined in the Materials and Methods. Relative differences in the cellular metabolite concentrations of sensitive versus resistant cells highlight the extent to which the acquired metallodrug resistance results in an altered metabolome (Figure 1). We observe that some responses, such as an increase in palmitoylcarnitine and a decrease in lactate upon resistance, are shared across the two metallo-resistance phenotypes. Nevertheless, many of the metabolic changes associated with resistance are drug-specific. Pyruvate and carnitine concentrations, for example, are higher in RuR cells but lower in OxR cells when compared to their sensitive counterparts. Palmitoyl-CoA, on the other hand, is lower in RuR cells and higher in OxR cells when compared to their parental sensitive counterparts (Figure 1).

With the aim of investigating whether the observed changes in cellular metabolite concentrations (Figure 1) translate to changes in metabolic flux, we integrated experimentally determined growth rates (Appendix A), intracellular metabolite concentrations (Figure 1) and exchange rates (Appendix A) in a genome-scale metabolic model (GSMM) of HCT116. We constrained four instances of the GSMM: an oxaliplatin-resistant (OxR) and a parental sensitive counterpart (sensitive), a BOLD-100/KP1339-resistant (RuR) and a parental sensitive counterpart for the DMSO-containing medium (sensitive-DMSO). Measuring 110 metabolite concentrations and 37 exchange fluxes, we constrained the solution space of a model with 6479 metabolites and 6716 reactions. Growth rates were used to constrain the biomass production of each model instance. Resistant cells grow slower than sensitive cells and OxR cells grow even slower than RuR cells (Appendix A). Exchange rates (Appendix A) were determined from time-course measurements of the medium composition and were applied as flux bounds on the corresponding import and export reactions of the model. Intracellular metabolite concentrations were applied as constraints using the pyTFA package [47]. Using a parsimonious thermodynamic flux variability analysis (pTFVA), as outlined in the Materials and Methods, we calculated flux solutions for each of the four model instances, each of which was constrained with the corresponding experimental data. By incorporating the growth and exchange rates as well as the intracellular metabolite concentrations into a GSMM, we were able to calculate possible changes in metabolic fluxes. Metabolic rates, rather than concentrations, could then be normalized according to the cellular growth rate observed under those conditions. We compared the four sets of flux solutions against one another, both before and after normalizing all flux values by the respective growth rate (Figure 2). Growth rate normalization was implemented by dividing all of the calculated flux values by the experimentally measured growth rate used to constrain that model instance.

The maximum relative standard error observed across the metabolite measurements was less than 15%. Thus, when integrating the data into the GSMM and comparing flux differences between condition-specific instances of the model, we used a cutoff of 15% to determine whether fluxes were significantly different across conditions. Comparing resistant cells to sensitive cells, we identify pathways with the most prominent changes in flux upon acquired resistance (Figure 2). Differences in flux observed prior to growth standardization directly correspond to predictions of in vivo fluxes. Differences in flux observed post growth standardization are no longer predictions of in vivo fluxes, but are predictions of flux differences that are assumed to be the direct result of a metabolic reprogramming upon acquired resistance rather than changes in growth rate.

Initially, the oxidative phosphorylation pathway shows the highest amount of flux changes in response to OxR. Upon growth normalizing, however, it is RuR that shows a higher number of flux changes in this pathway. Furthermore, what initially appears to be significant differences in flux through the cholesterol and lipid metabolism, largely disappears upon growth normalization. Changes in the subsystem reactive oxygen species (ROS) detoxification seem minimal prior to growth normalization; the normalized results, however, indicate significant changes in flux with regards to detoxification. While the number of reactions that appear to be affected in starch and sugar and tricarboxylic acid (TCA) metabolism appears to be drug resistance-specific prior to growth normalization, this effect disappears upon growth normalization. The comparison of non-normalized and growth-normalized results in Figure 2 emphasizes that observed changes in metabolite concentrations are not necessarily indicative of cellular changes in flux. It further highlights that flux results must be growth normalized in order to distinguish a resistance model effect from a growth effect when comparing the metabolic profiles of resistant and sensitive cells. Changes in the pentose phosphate pathway (PPP), oxidative phosphorylation, glycolysis/gluconeogenesis, TCA, nucleotide, ROS and fatty acid pathways, for example, appear to be a direct result of acquired resistance when comparing OxR and RuR to their parental sensitive counterparts (Figure 2).

### 3.2. Metallodrug Resistance Is Linked to Changes in Energy Metabolism

Integrating metabolite measurements into GSMMs allows for growth rate normalization of the calculated fluxes which in turn allows for a direct flux comparison between resistant and sensitive cells. The reprogramming of energy metabolism to support cell growth and proliferation is a major hallmark of cancer [1] and has previously been linked to the emergence of acquired drug resistance [3]. To further investigate the role of a reprogramming of energy metabolism upon acquired metallodrug resistance, we used the four instances of the HCT116 model (OxR, sensitive, RuR, sensitive-DMSO) to specifically assess differences in flux in pathways related to energy metabolism.

In the growth conditions considered here, glucose acts as the primary carbon source (Appendix A). Glucose is catabolized to pyruvate, generating two ATP during glycolysis. Pyruvate can then be transported into the mitochondria and converted to acetyl-CoA which then enters the TCA cycle or, in what is known as the Warburg effect in cancer cells [67], pyruvate can be converted to lactate. Acetyl-CoA can also be generated from fatty acid oxidation and sometimes amino acid catabolism (see [68] for a review). Fluxes corresponding to these three well-established energy pathways of colorectal cells along with the oxygen consumption are shown for each cell type in Figure 3. While glutaminolysis is another common means by which cancer cells support the Warburg effect [69], we did not measure high glutamine uptake rates in the considered growth conditions. In fact, our determined glucose and glutamine uptake rates are in the same orders of magnitude as previously determined for HCT116 cell lines grown in fetal bovine serum [61].

We observe that OxR cells convert less pyruvate into lactate, but in turn consume a higher relative amount of fatty acids compared to their parental sensitive cells. RuR cells, however, show a high glycolytic flux and a high oxygen consumption as well as higher fatty acid consumption than their sensitive counterparts (Figure 3). Notably, the flux values shown in Figure 3 are growth normalized and may therefore not directly correspond to what would be observed in a traditional oxygen consumption rate (OCR) versus extracellular acidification rate (ECAR) experiment [70]. When comparing experimentally determined OCR and ECAR measurements to the non-normalized model results, we find a close agreement with regard to the differences in glycolysis and respiration between sensitive and resistant cells (Appendix A); thus further validating the set model constraints.

With the four instances of the HCT116-specific GSMMs, further conditions encountered in the tumor environment can be simulated. Simulating the effect of hypoxic growth conditions, we first set the oxygen influx for each model instance to the minimum possible value and then observe the minimum required fatty acid influx as we iteratively increase the oxygen influx, thus plotting the growth normalized production envelope of oxygen versus minimum fatty acid influx for each of the four conditions (Figure 4). While RuR cells appear to have a lower tolerance for hypoxic conditions, they also have a higher fatty acid requirement under those conditions when compared to the sensitive simulations (Figure 4c,d). While the same difference can be observed between OxR and sensitive simulations, it is less pronounced (Figure 4a,b).

We then set a minimum possible fatty acid influx and iteratively increased the total fatty acid influx to the model while calculating the minimum required oxygen influx (Figure 4). We repeated this calculation for various biomass constraints and note that there is an optimal fatty acid influx for minimizing the total oxygen required. In fact, this optimal value corresponds directly to the fatty acid uptake rates observed in Figure 3d and is in accordance with the parsimonious thermodynamic flux variability analysis which minimizes the total sum of fluxes (see Section 2 for details).

Crucially, while a direct comparison between resistant and sensitive cells for each drug respectively can be made, we cannot make a direct comparison between the two drug resistance models (Figure 1, Figure 2, Figure 3, Figure 4 and Figure 5). Because RuR cells were grown in a DMSO-containing medium whereas OxR cells were not, we cannot, at this stage, distinguish a resistance model-specific effect from a solvent background-induced effect.

### 3.3. Growth Rate and Medium Normalization Allows for a Direct Comparison of Fluxes of Cells Grown across Heterogeneous Conditions

In order to be able to compare the metabolic profiles of the two metallodrug resistance phenotypes directly, we finally normalized the flux results obtained from the metallo-resistant model instances against their respective parental sensitive counterparts (see Section 2 for further details). By dividing growth rate normalized and feature-scaled flux values calculated for the resistant models by those calculated for the respective sensitive models, we add a further normalization step. This normalization step eliminates observed differences in flux values that are the result of differences due to the presence of DMSO-background. Because the parental sensitive counterparts were grown in the same medium as their resistant counterparts, we can assume that shared differences in flux between sensitive and resistant cells are the result of differences caused by DMSO. As such, this normalization step allows us to directly compare the two acquired resistances, OxR and RuR, to one another even though RuR, unlike OxR, was grown in a medium with low solvent (DMSO) background. The comparison of OxR and RuR (Figure 6) cells highlights an upregulation of fluxes associated with amino acid and fatty acid metabolism in RuR. OxR cells, on the other hand, show an upregulation in glycolysis and starch and sugar metabolism when compared to RuR cells (Figure 6b).

Notably, when comparing the OxR and RuR model instances to their respective parental HCT116 drug-sensitive counterparts, prior to growth normalization, we identified 1039 (OxR) and 1180 (RuR) fluxes that were significantly different. Upon growth normalization, these numbers reduced to 743 (OxR) and 883 (RuR), highlighting that hundreds of differences observed in the non-normalized results are simply the result of a difference in growth rate. The OxR versus RuR comparison upon growth-media DMSO-background normalization highlighted 670 different reactions, suggesting that another 73 of reactions were initially observed as significantly different because of presence of 0.5% DMSO.

## 4. Discussion

Genome-scale metabolic models (GSMMs) provide a platform for integrating omics data sets and for analysing them in the context of metabolic fluxes. As we have shown, GSMMs can be constrained using both extracellular and intracellular metabolite concentrations to study metallodrug resistance in colon cancer. Approximately one-hundred metabolite constraints were applied to study the effect of changes in their concentrations in thousands of reactions. Colorectal-specific GSMMs have previously been constructed [12,48,58,71] but have not yet been applied to study metallo-drug resistance specifically. Here, we compared metabolic flux alterations in HCT116 cell models with acquired oxaliplatin- (OxR) vs. BOLD-100/KP1339 resistance (RuR) relative to parental, drug-sensitive HCT116 cells grown in the respective growth media without or with 0.5% DMSO.

In this study, we investigated various pathways in silico including glycolysis, the tricarboxylic acid cycle, fatty acid and amino acid metabolism, beta-oxidation, the pentose phosphate pathway. A comprehensive stable isotope resolved metabolic flux analysis considering such a diverse set of pathways would require the application of multiple different positionally labelled isotopic tracers [27,72]. In addition to economic factors, practical challenges may also play a role in experimental design. The application of palmitate to study beta-oxidation, for example, requires the conjugation of fatty acid free bovine serum albumin [73]. Moreover, it usually has high background contamination from plastic materials [74]. Finally, stable isotope labeling in living organisms is even more complex from a data evaluation perspective [75]. Thus, a purely experimental study that provides a holistic analysis of metabolic reprogramming in cancer is currently infeasible.

There is no simple relationship between changes in metabolite concentrations and changes in flux [76]. This notion also applies to acquired resistance in the HCT116 colorectal cancer cell line. We have shown that observed differences in metabolite concentrations between resistant and sensitive conditions may not necessarily reflect a drug resistance-specific response but may instead arise as a result of differences in growth rate or solvent conditions. If we want to compare changes in metabolic flux of cells grown in heterogeneous conditions, data needs to be normalized in order for valid comparisons to be made. Here we have outlined a procedure for this kind of normalization based on thermodynamic genome-scale metabolic modelling of the HCT116 cell line.

Accurate comparative profiling of metabolic changes observed across heterogenous conditions remains a challenge. Differences in growth rates and impact of solvent necessities will result in observed differences in metabolite concentrations but are not causal to a reprogramming of metabolism [77]. Considering cellular fluxes as the metabolic phenotype through the use of GSMMs has the advantage that fluxes, unlike concentrations, can easily be normalized with regard to other rate measurements, such as growth rate or exchange rates (e.g., [78,79,80]).

As a prerequisite to produce accurate, quantitative metabolomics data [77], we normalized the metabolite amounts to total protein content, calculating absolute concentrations based on internal standardization. Even though this approach is superior to relative quantification and is able to compensate for technical variation of the sample preparation and differences in extracted biomass, it does not account for biological processes like growth or environmental factors such as differences in medium composition.

Integrating the metabolite data as part of a thermodynamic flux analysis allows us to normalize the calculated reaction rates by the growth rate observed under the corresponding conditions. We showed that growth-normalization reduces the number of reactions that are different between resistant and sensitive model instances and changes some of the conclusions about altered metabolic pathways entirely. Growth normalization is therefore a critical step when looking for drug-specific metabolic phenotypes. A second limitation to studying non-normalized metabolite concentrations is that data obtained from heterogeneous conditions cannot be directly compared. Using the growth normalized flux results we further normalized each resistant model against its sensitive counterpart which was grown in an identical medium and solvent composition. Hence, we were able to do a direct comparison between the two metallo-resistances and to identify drug resistance-specific responses.

A limitation to our approach is that we first normalized our flux results to differences in growth rate and then normalized each resistant model against its respective counterpart. This means that we are unable to capture emergent properties that result from both differences in growth rate and solvent impact; we assume that a combined effect of the two is minimal. Furthermore our flux analyses assume metabolism to be in steady-state, such that intracellular concentrations are constant. Nonetheless, we have clearly demonstrated that a growth and medium/solvent normalization is non-trivial as it allows for comparisons across heterogeneous conditions. We expect this method to be of wider applicability in studies where the effects of medium compositions, such as the availability of carbon sources to a cell, are of interest.

Time-dependent changes of metabolite profiles have previously been considered [81,82], but are not typically integrated at a genome-scale level. Measuring metabolite concentrations alongside cell counts at various time points and quantifying the relative metabolite abundance per cell using linear regression, Dubuis et al. [81] account for deviations from steady-state. The method was then further developed, using intermediates of fatty acid metabolism and other metabolites to account for differences in cell size [82]. While the difference in cell size can be interpreted as a proxy of growth rate, it cannot be assumed that the observed changes in metabolite concentrations directly translate to differences in metabolic activity, i.e., fluxes. Metabolic responses associated with an acquired metallodrug resistance in cancer have not yet been studied extensively using constraint-based flux analyses [39,83,84]. The use of GSMMs to integrate metabolomics data to study cellular fluxes, however, provides multiple new opportunities in this field.

Defense mechanisms and acquired resistance are well known phenomena when applying metal-based drugs as anticancer agents. Reduced efficacy due to acquired resistance remains a major challenge in systemic anticancer therapy. The complexity is increasingly recognized, as the contribution of epigenetic and metabolic effects will be uncovered. Drug-specific and tumor tissue specific mechanisms have been described, and more recently the tumor microenvironment has come into focus [85]. Accordingly, response profiling with metabolomics analysis can be a powerful tool for investigating drugs and drug candidates [28,29] and dissecting emerging resistance [86]. Currently, only a handful of studies consider metallodrugs applied to cancers with metabolomics [87,88,89], and even fewer investigate acquired metallodrug resistance [53].

In this work, we consider an in vitro study of colon cancer. Gastrointestinal cancer cell lines, including colorectal cancer cell line HCT116 activate beta-oxidation as a response to oxaliplatin treatment and conversely become more sensitive to oxaliplatin upon inhibition of fatty acid catabolism [23]. A seminal study in the field integrates both metabolomics and transcriptomics and finds that, within 59 NCI60 cell lines, the metabolic basis of platinum-sensitivity can largely be attributed to energy metabolism (TCA cycle, glutaminolysis, pyruvate metabolism), lipoprotein uptake, and nucleotide synthesis [90]. The results from our in silico analysis are in line with these findings, also highlighting the importance of energy metabolism (OXPHOS, glycolysis, TCA).

Figure 6 highlights the relevance of fatty acid metabolism, as fluxes from this subsystem contribute to 12.5% of all observed differences (excluding all transport reactions) between the RuR and OxR, showing elevated fluxes in RuR. This supports existing evidence of beta-oxidation activation in response to metallodrug treatment [23]. Interestingly, for the majority of observed flux differences between OxR and RuR, flux values are higher in the RuR, implying a higher metabolic activity in this phenotype, independently of differences in growth rate. OxR, however, does exhibit elevated activity in the glycolysis/glyconeogenesis and pentose phosphate pathways. Thus, our comparison of the growth rate- and medium-normalized reaction rates supports the notion that an acquired resistance to the two metallodrugs is marked by differences in their metabolic phenotype with an overall higher metabolic activity in the RuR system.

The high metabolic plasticity of cancer cells enables efficient detoxification and protection strategies [85]. Normalization of the flux values by growth rate substantially reduces the observed differences (Figure 2) in most of the investigated pathways. In contrast, both the pentose phosphate pathway and ROS detoxification subsystem, which includes glutathione-synthesis, were emphasized to the same extent in both the OxR and RuR resistance models upon growth standardization. This supports the notion that metallodrugs interfere with cellular redox homeostasis and stimulate a readiness to counter reactive oxygen species (also by synthesizing NADPH via the pentose phosphate pathway) which has previously been described to conjugate glutathione to platinum complexes with glutathione-S-transferase [91].

Despite shared commonalities like the production of ROS, it is expected that RuR and OxR models display different metabolic phenotypes, because of known differences in their modes of action [26,92,93]. Oxaliplatin, for example, is primarily a DNA targeting drug, whereas BOLD-100/KP1339 has recently been found to have a prodrug nature and is capable of causing ER-stress and the downregulation of GRP78, encoding an endoplasmic reticulum chaperone protein, which has been linked to malignancy [94]. It is widely accepted that DNA repair mechanisms play a crucial role in resistance to oxaliplatin [95]. It is important to note that, using GSMM, we have here focused solely on metabolic changes to compare metabolic reprogramming of the two acquired resistances but cannot exclude further regulatory events.

The comparison of fluxes through key energy metabolism reactions (Figure 3) shows that both acquired resistances are defined by lowered glycolytic flux than their sensitive parental cells, although this is less pronounced with RuR. Growth normalization does not affect this observation (Appendix A). The same cannot be said about the fatty acid beta-oxidation, where upon growth standardization the acquired resistance models both show a higher fatty acid requirement than their sensitive controls (Figure 3d; Appendix A). Additionally, upon growth normalization OxR has lower and RuR higher respiration rates than corresponding sensitive counterparts (Figure 3a). The calculated rates correspond well to the experimentally determined results with a Seahorse assay (Appendix A). As expected the experimentally determined and non-normalized in vitro results align more closely with the non-normalized flux values modelled in silico.

Drastic changes in oxygen and fatty acid availability are known stress conditions in a tumor microenvironment, and are assumed to be managed with metabolic adaptations [4]. Lipid dependency, for example, is more pronounced under hypoxic conditions and relies on the uptake of extracellular fatty acids [73,96,97]. We thus used the condition-specific instances of our constrained GSMM to further inspect the relationship between hypoxia and fatty acid uptake. We found that the composition of fatty acids taken up changes in response to oxygen limitation (Figure 4). Under normoxic conditions linolenate can act as the sole fatty acid source. As oxygen limitation becomes more pronounced, linoleate, arachidonate, oleate, stearate and finally palmitate are also required. RuR cells require less fatty acids under oxygen limitation compared to their sensitive counterpart (Figure 4c,d); while the same is true for OxR, the observed difference is notably less pronounced (Figure 4a,b).

Additionally, the investigation of minimum oxygen requirement at various fatty acid influxes (Figure 5) revealed that the optimal fatty acid composition, which has the lowest oxygen demand, is the same across growth rates. Overall, OxR has the lowest oxygen requirement, which suggests that if sufficient fatty acids are available, OxR will be the most resilient of the investigated models against hypoxia (Figure 5).

## 5. Conclusions

There are different ways to capture the metabolic phenotype of a cell. Metabolic profiling via metabolomics provides an interrogation window of the intracellular concentrations at a given point in time. Extracellular concentrations measured over time provide insight to the cellular uptake and excretion rates of cells. Together they can be integrated to constrain the solution space of a genome-scale metabolic model. The calculated flux values can then be normalized according to growth rates and environmental conditions, allowing for drug resistance specific metabolic responses to be identified across heterogenous conditions. We find the outlined normalization steps to be crucial in the interpretation of the results and show that metabolic reprogramming is more extensive in BOLD-100/KP1339 resistant cells than in oxaliplatin resistant cells. We identify pathways, such as fatty acid and amino acid metabolism, to be upregulated in response to a resistance acquired to a ruthenium-based drug when compared to a platinum-based drug. All in all, genome-scale metabolic modelling provides a valuable platform for putting observed changes in metabolite concentrations in the context of metabolic fluxes. 

## Figures and Tables

**Figure 1 cancers-13-04130-f001:**
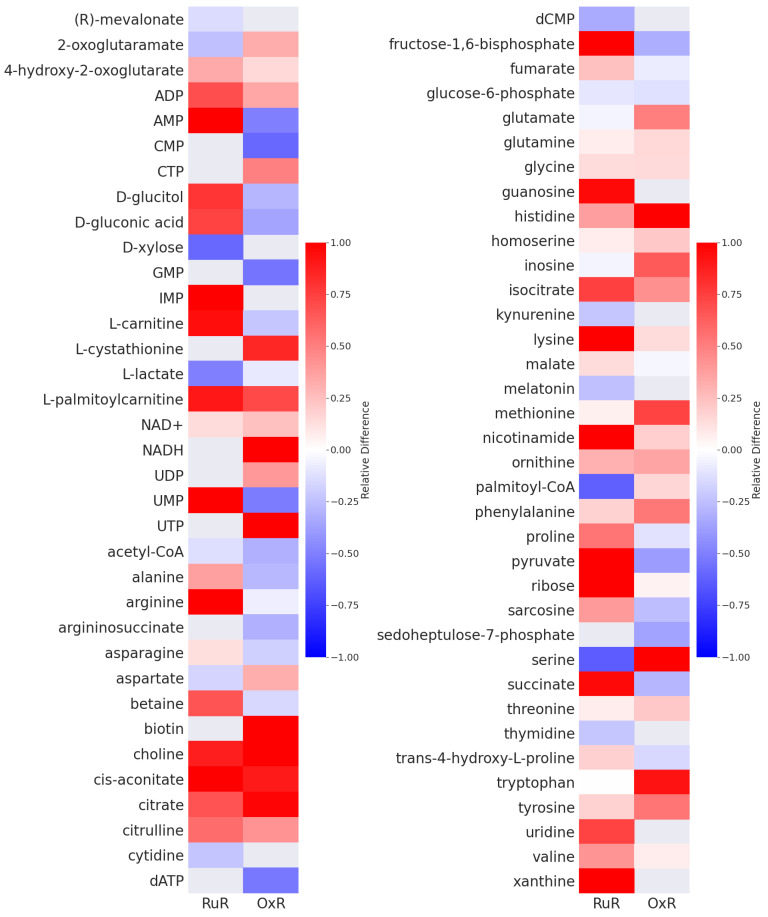
Observed changes in metabolite concentrations upon acquired resistance. Oxaliplatin (OxR) and BOLD-100/KP1339 (RuR) resistance models are compared to their sensitive counterparts. Relative differences in the measured metabolite concentrations of resistant and sensitive cells are shown. A positive relative difference (red) indicates a higher metabolite concentration in the resistant as compared to the parental sensitive model, whereas a negative relative difference (blue) indicates a lower metabolite concentration in the resistant than in the sensitive counterpart. Relative differences were calculated from using the mean values of six replicates. Only those metabolites for which we observed an absolute relative change greater than 15% between sensitive and resistant, in at least one of the two conditions, are shown. The following abbreviations were used: ADP—adenosine diphosphate, AMP—adenosine monophosphate, CMP—cytidine monophosphate, CTP—cytidine triphosphate, GMP—guanosine monophosphate, IMP—inosine monophosphate, NAD—nicotinamide adenine dinucleotide, UDP—uridine diphosphate, UMP—uridine monophosphate, UTP—uridine triphosphate, CoA—coenzyme A, dATP—deoxyadenosine triphosphate, dCMP—deoxycytidine monophosphate.

**Figure 2 cancers-13-04130-f002:**
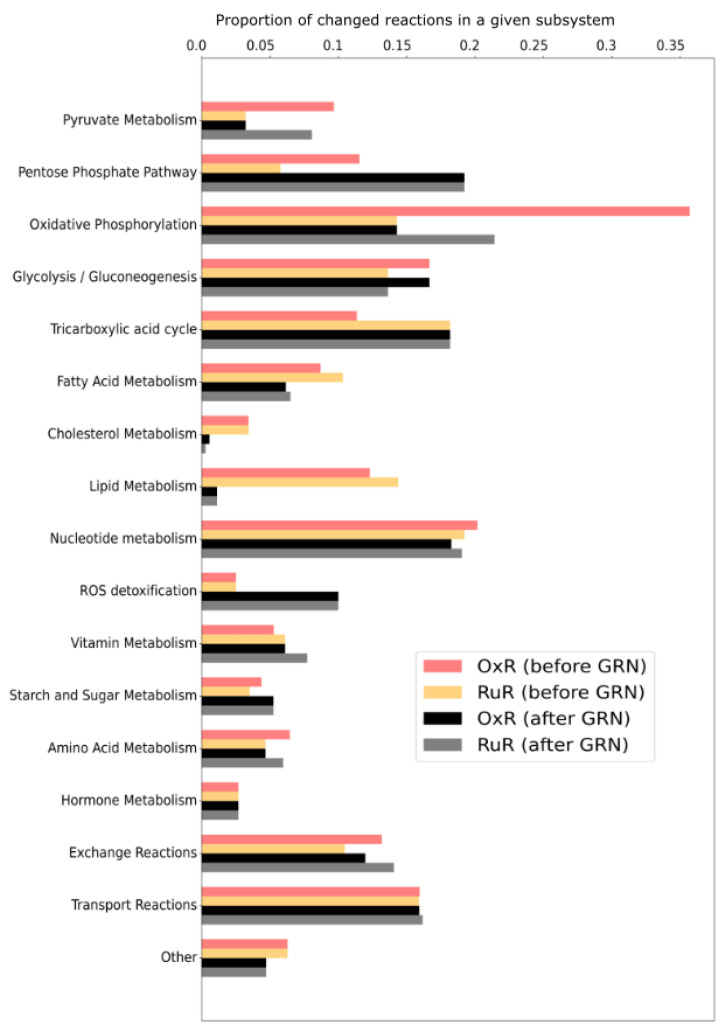
Metabolic fluxes in resistant versus sensitive models before and after growth rate normalization. Both intracellular and extracellular metabolite constraints were applied to generate four instances of the HCT-specific GSMM [oxaliplatin (OxR) and BOLD-100/KP1339 (RuR) and their parental sensitive counterparts (sensitive and sensitive-DMSO, respectively)] as described in the Section 2. A parsimonious thermodynamic flux variability analysis (pTFVA) was done on each model instance. Flux values of the resistant instances were compared to their respective controls. Metabolic reactions that had an absolute relative difference greater than 15% in both the highest possible and the lowest possible flux value were considered to be different. The proportion of reactions that show a difference in flux between the OxR and sensitive condition [OxR before growth rate normalization (GRN); red bars] and the RuR and sensitive conditions (RuR before GRN; orange bars) are shown for each subsystem (pre-defined pathways in the GSMM). All flux values were then normalized according to the corresponding growth rate of that condition (Appendix A) and were again checked for a relative difference between OxR (OxR after GRN; black bars) and RuR (RuR after GRN; gray bars) and their sensitive controls. Subsystems for which no relative changes in flux between resistant and sensitive instances were observed were omitted from the figure for clarity.

**Figure 3 cancers-13-04130-f003:**
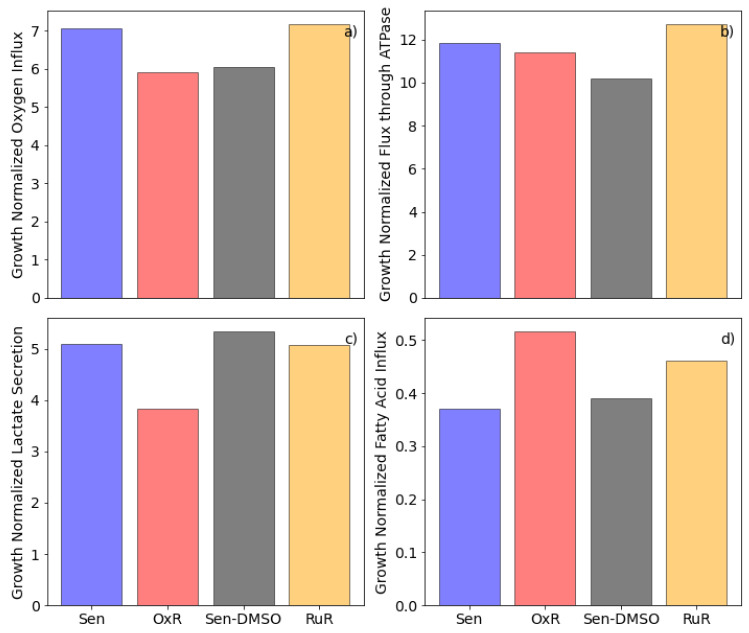
Comparison of fluxes through key energy metabolism reactions for ruthenium- and oxaliplatin-based resistant cells after growth rate normalization. Growth normalized flux values through (**a**) the oxygen uptake reaction—HMR_9048, (**b**) the ATP synthase reaction—HMR_6916, (**c**) the lactate secretion reaction—HMR_9135, and (**d**) the fatty acid influx—sum of m01362s_FAx, m02387s_FAx, m02389s_FAx, m02646s_FAx, m02674s_FAx, m02938s_FAx, across the four model instances (sensitive, Sen—blue bars; sensitive in a DMSO-containing medium, Sen-DMSO—gray bars; oxaliplatin-resistant, OxR—red bards; BOLD-100/KP1339, RuR—yellow bars) are shown. Fatty acid influx is the combined influx of stearate, palmitate, oleate, linolenate, linoleate, arachidonate.

**Figure 4 cancers-13-04130-f004:**
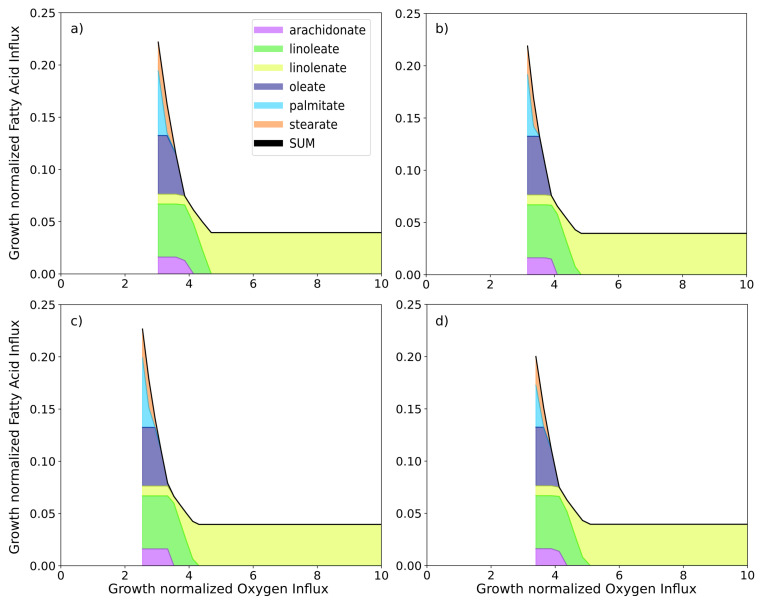
Simulating minimum fatty acid uptake requirement in response to various oxygen uptake constraints. Oxygen influx constraints were applied to each of the model instances (**a**) sensitive, (**b**) oxaliplatin-resistant, (**c**) sensitive in DMSO-containing medium, and (**d**) BOLD-100/KP1339-resistant. Using a parsimonious thermodynamic flux variability analysis (pTFVA; see Methods and Materials for details) the minimum possible fatty acid uptake was calculated for each oxygen constraint as shown. Fatty acid influx is the combined influx (black line) of stearate (orange), palmitate (cyan), oleate (dark purple), linolenate (yellow), linoleate (green), arachidonate (light purple).

**Figure 5 cancers-13-04130-f005:**
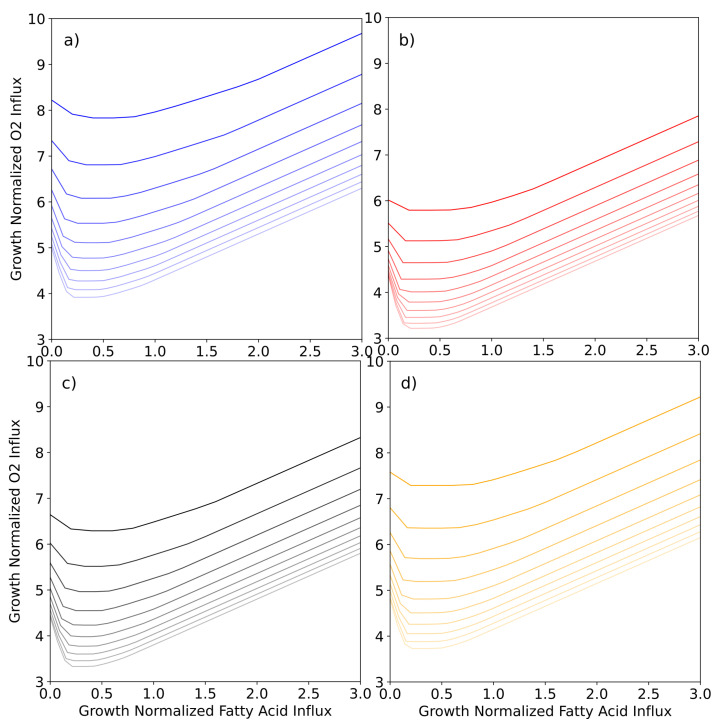
Simulating minimum oxygen requirement in response to various fatty acid uptake and biomass constraints. Fatty acid influx constraints were applied to each of the model instances (**a**) sensitive—blue lines, (**b**) oxaliplatin-resistant—red lines, (**c**) sensitive in DMSO-containing medium—black lines, and (**d**) BOLD-100/KP1339-resistant—yellow lines. Using a parsimonious flux variability analysis (pTFVA; see Methods and Materials for details) the minimum possible oxygen uptake was calculated for each fatty acid constraint. The calculations were performed for various growth rate constraints ranging from 5 to 15 h^−1^, as indicated by the fading lines. Fatty acid influx constraints were applied as the combined influx of stearate, palmitate, oleate, linolenate, linoleate, arachidonate.

**Figure 6 cancers-13-04130-f006:**
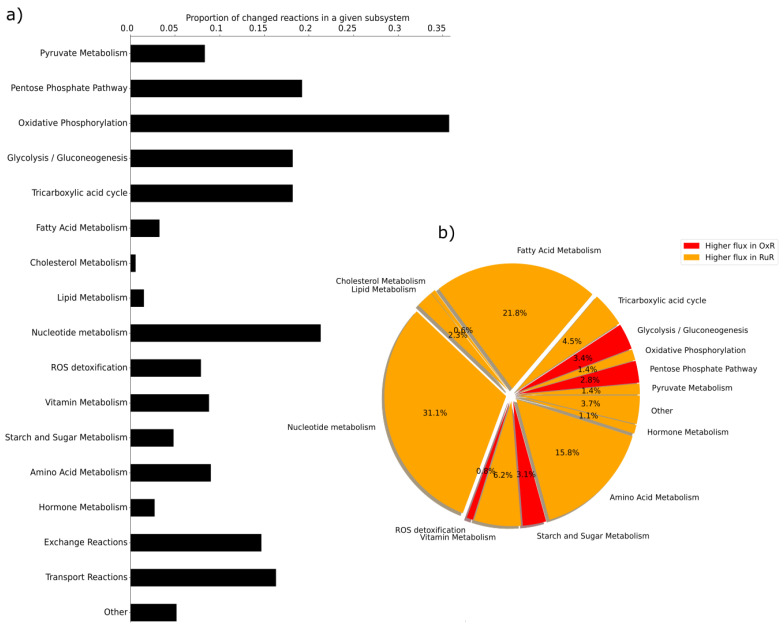
Predicted differences in the metabolic fluxes of ruthenium- and oxaliplatin-based resistances after growth rate and medium normalization. Both intracellular and extracellular metabolite constraints were applied to generate four instances of the HCT-specific GSMMs (oxaliplatin (OxR) and BOLD-100/KP1339 (RuR) and their sensitive parental counterparts (WT and WT-DMSO, respectively)) as described in the Materials and Methods. A parsimonious thermodynamic flux variability analysis (pTFVA) was done on each model instance. Each set of flux values was divided by the corresponding flux through biomass (Appendix A), thus normalizing for differences in growth. Flux values were then feature-scaled to lie between 0 and 1 and the flux values obtained in the drug-resistant instances were divided by the flux values of the corresponding control instances, thus normalizing for difference in medium composition. (**a**) The relative changes in flux between OxR and RuR instances were calculated and the total number of reactions that showed an absolute relative difference greater than 15% in relative upper and lower flux values were counted for each subsystem. The proportion of reactions that are significantly different in each subsystem is shown as black bars. Subsystems for which no relative changes in flux between the two resistant instances were observed were omitted from the figure for clarity. (**b**) Percentage of reactions in a subsystem which were identified as significantly different out of all reactions that were identified as significantly different between the two conditions are shown as a pie chart. Subsystems for which the total flux values were higher in OxR are shown in red. Subsystems for which total flux values were higher in the RuR are shown in orange. The subsystems transport reactions were omitted from this analysis as together they make up over 95% of the significantly different reactions.

## Data Availability

All data and code used to conduct the analyses presented in this manuscript are available on GitHub (https://github.com/HAHerrmann/Hct116_DrugRes, accessed on 12 August 2021) and Zenodo (DOI: 10.5281/zenodo.4633725 (https://zenodo.org/record/4633725, accessed on 12 August 2021)). Metabolomics data (LC high-resolution mass spectrometry-based metabolomics dataset in rawdata and total protein contents corresponding to the samples) have been deposited to the EMBL-EBI MetaboLights database [98] with the identifier MTBLS2665 for the OxR-batch and MTBLS2681 for the RuR-batch. The complete dataset can be accessed at https://www.ebi.ac.uk/metabolights/MTBLS2665, accessed on 12 August 2021 and https://www.ebi.ac.uk/metabolights/MTBLS2681, accessed on 12 August 2021 for the OxR- and RuR-batch, respectively.

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
