# Peer review of "Thermodynamic Genome-Scale Metabolic Modeling of Metallodrug Resistance in Colorectal Cancer"

_cancers, 2021, doi:10.3390/cancers13164130_

Round 1

Reviewer 1 Report

The authors investigate the relation of drug resistance and metabolic fluxes by thermodynamically constrained pFBA in chemoresistant HCT116 cells. Intracellular concentrations were used to thermodynamically constrain the directionality of intracellular reactions. Extracellular concentrations were used to compute uptake and excretion rates and constrain the exchange reactions of the GSMMs. The authors show that normalization for growth is essential to distinguish growth effects from metabolic reprogramming effects. This is a simple normalization, yet the implications for accurate analysis of drug resistance are very relevant, and would also be of interest for analysis of other metabolic flux studies. However, it is not clear how parts of the methods and results are obtained and there are methodologically concerns as follows.

A better description of the methods is required.

In the results section, I find “the mean value of three replicates”, “the mean value of four replicates”, and “the mean value of six replicates”. However, I do not find the mention of replicates at the methods section at all.

Metabolite concentrations are obtained at 0, 24, 48, and 72h. Is TFBA performed at each time point? Are averages used? Or at a selection of time-point(s)?

Page 6 line 259: “We then scaled our metabolite concentrations to fall within that same range”. Why do quantitative metabolite measurements have to be scaled?

Figure 1 shows intracellular metabolite differences. What is the percentage of changed metabolites compared to the total number of measured metabolites? I could not access the source data.

On page 3, the authors mention that pFBA may reach infeasible solutions, whereas thermodynamic FBA ensures thermodynamically valid fluxes. In this work, what is the actual contribution of the changed metabolites in figure 1 to the thermodynamics FBA? Which reactions are thermodynamically constrained by the intracellular concentrations? To what extend is the flux space constrained by thermodynamics of (intracellular) reactions?

Next to the measured intracellular metabolites, it would be interesting to show a (supplementary) figure with the changes in exchange rates, in particular because these were amongst the changed subsystems in figure 2.

In figure 2, what pathway definition is used? At the title, I would say “Proportion of changed reactions in a given subsystem”.

Figure A2 shows the fitted and measured growth. It is a little hard to read the figure because of the scale. I suggest to zoom in the figure up to 75 hours. In addition, does exponential growth after day 3 make sense when the lag phase may be reached?

Page 5 and figure A3. Exchange rates are calculated per initial biomass, based on the measured cell dry weight, if I understand the methods correctly. The error bars (standard errors) on the dry weight are large, up to 30%, in figure A3. Moreover, Sen-DMSO is not shown. This figure raises several questions.

Could the individual points be shown as well in this graph?

Is the observed dry weight per cell expected to be larger in resistant cells compared with sensitive cells?

What is the effect of this greater dry weight and high uncertainty on the downstream analyses? Since normalization per growth rate is the important point in this paper, the division of exchange rates by the cell dry with per cell is as well. How robust is the growth normalization considering these differences and errors?

Reviewer 2 Report

  • Interesting study and I enjoyed reading this investigation. My only concern is about the Abstract section that could be more informative and the authors might add more results.
  • There are some studies that showed metabolic heterogeneity in different cancers using GSMM (https://www.mdpi.com/2075-4426/11/6/496), so, it would be nice if the authors mention this at the beginning of the Abstract section that we need a personalized(customized) treatment for cancers regarding this heterogeneity.
  •  At the beginning (in line 28), might be replaced by a more relevant word.
